# Pitting Corrosion Behavior and Surface Microstructure of Copper Strips When Rolled with Oil-in-Water Emulsions

**DOI:** 10.3390/ma14247911

**Published:** 2021-12-20

**Authors:** Xudong Yan, Jianlin Sun

**Affiliations:** 1School of Materials Science and Engineering, Beihang University, No. 37 Xueyuan Road, Beijing 100191, China; 2School of Materials Science and Engineering, University of Science and Technology Beijing, Beijing 100083, China; sjl@ustb.edu.cn

**Keywords:** copper strip, rolling reduction, surface microstructure, emulsion, pitting corrosion

## Abstract

Copper strips experience severe corrosion when rolled with an oil-in-water (O/W) emulsions lubricant. The effects of rolling reduction on the pitting corrosion behavior and surface microstructure of Cu strips were studied in detail using electrochemical measurements and electron back scattered diffraction (EBSD) analysis. It was found that the corrosion current densities of the rolled Cu strips increased with accumulated reduction, which also lowered the pitting potentials and weakened their corrosion resistances. Therefore, the corrosive tendency of Cu strips under different rolling reductions (ε) followed the order of ε_0%_ < ε_20.7%_ < ε_50.6%_ < ε_77.3%_. The Cu surface easily reacted with chlorine, sulfur, and carbon components from O/W emulsions to generate pitting corrosion. Under the interactive effect of pitting corrosion and stress corrosion, pits expanded along the rolling direction. The aggregation of anions in surface defects, such as dislocations, metastable pits, and microcracks, further accelerated the pitting corrosion of the surface.

## 1. Introduction

Copper (Cu) and its alloy strips have been widely applied in the fields of mechanical, aviation, and electronic information due to their excellent electrical conductivity, thermal conductivity, reliability, and workability [1,2,3,4,5,6]. Cold rolling is one of the key techniques of the Cu strip manufacturing processes. Oil-in-water (O/W) emulsions are commonly served as the technical lubricants to optimize the dimensional accuracy and to improve the surface quality of the Cu strips during cold rolling [7]. Although Cu has good corrosion resistance in an ordinary environment (dry atmosphere), it still suffers severe corrosion when exposed to high temperature, high humidity, cold, shock, vibration, and high shear stress environments during employment [8]. In particular, when Cu products are exposed to solutions such as O/W emulsions with complicated components for a long time, their surface qualities and usage performance will be weakened. This issue has attracted considerable attention from researchers [9,10].

In our previous studies, it was found that copper corrosion possessed an incubation period. Cu could react with O atoms in emulsions and gradually transformed to Cu^+^ and Cu^2+^. Then, components that contained hydroxide and carboxylate anions would adsorb on the Cu surface to generate copper compounds such as Cu_2_O, CuO, Cu(OH)_2_, CuCO_3_, and Cu_2_(OH)_2_CuCO_3_ [11]. Furthermore, pH values, water hardness, additives, and other auxiliary reagents influenced the corrosion properties of the Cu surface. Soluble ions in hard tap water changed the electrochemical conductivity, resulting in the disruption of the passive layers, which accelerated the pitting corrosion, and the weak alkaline solutions contributed to the inhibition of pitting corrosion [12]. However, the above studies only concerned the influences of properties of O/W emulsion itself on Cu corrosions. Cold rolling is a complicated process. With the rolling reduction accumulated, cold working deformations of Cu samples might develop in large evolutions on their grain boundaries and grain size [13]. These deformations not only affected the microstructure of Cu, but also had a significant impact on its corrosion behavior. 

Related scholars have studied the relationship between the deformation degree and corrosion behavior of copper, but the conclusions are not consistent. Robin [14] investigated the corrosion resistance of Cu grains in H_2_SO_4_ solutions during the cold-swaged and cold-wiredrawn machining process, and found the corrosion resistance decreased with the deformation degree. Lapeire [15] studied the influence of grain size on the electrochemical behavior of pure copper in 0.1 M HCl and pointed out that for a smaller grain size, a lower corrosion potential and higher corrosion current density were observed. On the contrary, Deng and Nikfahm [16,17] stated that the ultrafine-grained copper enhanced its anti-corrosion behavior. During the cold rolling process, although Cu grain has not experienced recrystallization, the atomic arrangement and surface defect distribution on the microstructure of Cu will be changed by rolling reduction, in which the corrosion properties were still unclear. Furthermore, with the rolling reduction accumulated, the distribution of residual stress on the Cu surface may also have an impact on its corrosive order. Therefore, a deep understanding of these aspects, including electrochemical processes, surface morphologies, microstructure evolutions, and mechanical properties on the corrosion behavior of the cold-rolled Cu strips need to be studied.

In this paper, the electrochemical performances of the rolled Cu strips as electrodes in O/W emulsions were investigated using potendiodynamic polarization and electrochemical impedance spectroscopy (EIS) measurements. The microstructure of Cu strips under different rolling reductions was observed by electron back scattered diffraction (EBSD) analysis. Their surface mechanical properties were also figured out and the pitting corrosion mechanism of Cu surface was discussed based on surface and cross-section characterizations. 

## 2. Materials and Methods

### 2.1. Materials and O/W Emulsions Preparation

The component of experimental Cu sample is shown in Table 1, which was obtained from the same raw strip as our previous study [18]. The strip was at hard state without any heat treatment and was cut with a size of 10 mm × 10 mm × 1.98 mm. Ethanol and de-ionized water were used for cleaning the Cu sample before the tests. 

The O/W emulsion was composed of 95 wt.% tap water and 5 wt.% emulsified oils. Mineral oil D130 and rapeseed oil are the main components of emulsified oils. Their composition and physiochemical properties are shown in Table 2. The other reagents included surfactants (sorbitan oleate); emulsifiers (oleic acid and triethanolamine), and anti-wear additives (Dibutyl Phosphite). Sorbitan oleate (C_24_H_44_O_6_, ≥98%, AR), oleic acid (C_8_H_17_CH = CH(CH_2_)_7_COOH, ≥99%, AR) and triethanolamine (N(CH_2_CH_2_OH)_3_, ≥99%, AR) were purchased from Sinopharm Chemical Reagent Beijing Co., Ltd., Beijing, China. Dibutyl phosphite ((C_4_H_90_)_2_POH, ≥99%, AR) was provided by Qianyang Technology Hangzhou Co., Ltd., Hangzhou, China. All chemicals were newly produced and used as received without further purification.

The preparation process of O/W emulsions is presented in Figure 1. Firstly, 1.4 g rapeseed oil, 2.1 g surfactants, 0.6 g emulsifiers, and 0.65 g anti-wear additives were gradually dissolved into 15.25 g base oil and mixed at 25 °C for 10 min to obtain the equably emulsified oils. Subsequently, the mixture was heated at 60 °C with a reactor and stirred for 20 min. It was then diluted with tap water to 5 wt.% at the same temperature. After stirring and cooling, and the O/W emulsion was completely prepared. The ionic concentration compositions of O/W emulsions were measured using ion chromatography, and the result is shown in Table 3.

### 2.2. Cold Rolling Tests

Cold rolling test of the Cu strip was conducted using a Φ130 × 220 mm two-high mill with a velocity of 13 r/min and a rolling power of 5.5 kW at 25 °C. O/W emulsions that served as lubricants were added at the deformation zone to obtain thinner strip with higher surface quality. The reduction ratio of each pass was constantly restricted at 20%. The rolled thickness of each pass is shown in Table 4. Due to the influence of the spring-back error, the actual strip thickness of each pass was larger. In this work, Cu strips under a different accumulated reduction of 0% (raw strip), 20.7% (small reduction strip), 58.6% (moderate reduction strip), and 77.3% (large reduction strip) were recollected as samples for subsequent electrochemical measurements and EBSD analysis.

### 2.3. Electrochemical Measurement

Electrochemical measurements were performed by a multichannel potentiodynamic system (VersaSTAT, AMETEK, Berwyn, PA, USA) equipped with a conventional three-electrode cell. A 4 cm^2^ platinum sheet was utilized as the auxiliary electrode, a saturated calomel electrode (SCE) functioned as the reference electrode, and the cold-rolled Cu strips were used as the working electrodes. The working electrodes were connected with Cu wire and embedded into epoxy resin, leaving a 1 cm^2^ cross-sectional area exposure for electrochemical experiments. All of the Cu electrodes were abraded with 800 to 2000 grit silicon carbide papers gradually and then polished with 0.5 SiO_2_ anti-scuffing paste to obtain a minor-like appearance before the electrochemical measurements. 

The open circuit potential (OCP) was first measured to achieve a steady state. Subsequently, EIS tests were performed at a frequency ranging from 10^5^ to 10^−2^ Hz, using an AC signal with an amplitude of 10 mV. The EIS data were analyzed carefully by Zsimpwin software (3.30d, Echem, Ann Arbor, MI, USA). Finally, potentiodynamic polarization measurements were carried out and the results were recorded from a potential ranging from −1.5 V to 1.5 V at a scan rate of 1 mV/s. Before the tests, the electrodes were immersed in the same O/W emulsion with a pH value of 7.8, which served as the corrosive medium. All of the electrochemical experiments performed referred to the OCP in a temperature-controlled water bath at 25 ± 2 °C for 2 h. All of the measurements were performed thrice to ensure a satisfactory reproducibility.

### 2.4. Surface Observations

The surface morphologies of the corroded Cu electrodes were observed with laser scanning confocal microscopy (LSCM, LEXT OLS4100, OLYMPUS, Tokyo, Japan) and field emission scanning electron microscopy (FE-SEM, Nova Nano-SEM450, FEI, Hillsboro, OR, USA). X-ray photoelectron spectroscopy (XPS, ESCALAB 250 Xi, Thermo Fisher Scientific, Bedford, MA, USA) was carried out to analyze the components of the corrosion products on the surface. The cross-section microstructure of the pits was analyzed by the combined use of FIB/SEM (Helios Nanolab 600i, FEI, Hillsboro, OR, USA). The dislocation configurations on the microstructure of the rolled Cu strips were observed by transmission electron microscopy (TEM, FEI Tecnai G2 F20, FEI, Hillsboro, OR, USA).

### 2.5. EBSD Characterizations

For the preparation of the EBSD samples, these four Cu samples were electropolished using a voltage of 3.0 V for 40 s in a phosphoric acid electrolyte, which consisted of 175 mL phosphoric acid and 825 mL deionized water. The microstructure of the Cu sample was examined in a JSM-7900F FE-SEM equipped with an EDAX OIM EBSD system and the operation voltage was set as 20 keV. The characterizations were conducted by scanning a large area of 500 × 500 μm^2^ with a step size of 0.8 μm to ensure that the results presented the real microstructure. All of the EBSD samples were placed with one accord, exposing the rolling directions (RD) and the transverse direction (TD) for microstructural observations. OIM Analysis software (6.2, EDAX, Philadelphia, PA, USA) was used for the EBSD data processing.

### 2.6. Mechanical Properties Tests

The microhardness of the near-pit sites on the Cu strips was implemented through an EM-1500L Vickers indentation instrument. As shown in Figure 2, under the effect of indentation load, the material began to crack at the top corner and the crack length could be measured when it was unloaded. The hardness (*H*) was calculated as follows,
(1)$$H=\frac{P}{S}=\frac{P\times 2\sin\frac{\theta }{2}}{d^{2}}=1.854\times \frac{P}{d^{2}}$$
where *P* is the preload; *d* is the diagonal length of indentation; and *θ* is 136°, representing the angle between the diamond indenter and sample plane.

Subsequently, referring to Lawn’s theory, two mechanical properties, namely the fracture toughness (*K*_IC_) and residual stress (*σ*_γ_), in Cu strips were deduced from the following data. Assuming the influence of *σ*_γ_ was uniform, *K*_IC_ could be calculated using Formula (2) [19],
(2)$$K_{IC}=\chi \frac{p}{C^{2/3}}+\frac{2}{\sqrt{\pi }}\sigma_{\gamma }C^{1/2}-\frac{2}{\sqrt{\pi }}\sigma_{\gamma }\frac{t}{C^{1/2}}$$
*t* and *χ* are the parameters that were further obtained by Formulas (3) and (4),
(3)$$t=\frac{d}{2\tan\theta }$$
(4)$$\chi =\delta {\left(\frac{E}{H}\right)}^{1/2}$$
where *δ* is the indenter geometry factor, *E* is the elastic modulus of Cu strips, and *C* is the crack length.

## 3. Results and Discussion

### 3.1. Electrochemical Corrosion Properties

Table 5 shows the detail electrochemical parameters, including the corrosion potential (*E*_corr_), corrosion current density (*i*_corr_), anodic Tafel slope (*β*_a_), and cathodic Tafel slope (*β*c), which are deduced through the Tafel linear extrapolation method (as shown in Figure 3a). It is found that the initial anodic current density linearly increases with the potentials. When the Cu electrode passes the Tafel polarization region, a “potential plateau” phenomenon is observed at this stage and the log*i*_corr_ value hardly moves with the increase of *E*_corr_. This “potential plateau” commonly referred to as a passivation region, which is due to the formation of intermediate species such as CuO_2_ and $\mathrm{Cu}{\mathrm{Cl}}_{2}^{-}$ on the electrode surface via the chemical reaction between Cu electrodes and solutions [20,21]. When the potential reaches a certain value, log*i*_corr_ increases again and the corrosion of the Cu electrode is accelerated. This generally causes pitting corrosion on the electrode surface and the potential value in this case is defined as the pitting potential (*E*_p_) [22]. 

Potentiodynamic polarization curves of Cu electrodes under different cold rolling reduction in the O/W emulsions are shown in Figure 3b. It is observed that there are not significant changes for both of the anodic slope (*β*_a_) and cathodic slope (*β*_c_) reflecting the similar polarization behavior of these four curves. The *i*_corr_ of the raw Cu electrode in the O/W emulsion is 1.23 × 10^−6^ A/cm^2^. With the accumulation of the rolling reduction, the values of log*i*_corr_ gradually increase and *E*_corr_ shifts towards being more negative in the anodic region. This indicates Cu suffers more serious corrosion as the rolling process extends. In particular, when the Cu strip experienced a 77.3% reduction, the *E*_p_ of the Cu electrode abruptly decreases and the passivation region subsequently disappears, which means that pitting corrosion occurs easily on the Cu surface with a larger reduction.

The Nyquist plots of the Cu electrodes in the O/W emulsions are displayed in Figure 4a. The curves show depressed semicircles in the high-frequency region, followed by straight lines in the low-frequency region. Generally, the semicircles are related to charge transfer resistance and double-layer capacitance. The low-frequency impedances are ascribed to Warburg impedances, which can be explained by the mass diffusion of the corrosion reactants and products towards or away from the Cu surface [23,24,25]. At the high-frequency impedance region, the electrode of the raw Cu strip exhibits the largest impedance and its diameter reduces with the increase of cold rolling reduction. Affected by O/W emulsion, Cu bulks are adsorbed by some soluble ions, then the film resistance and film capacitance are existed on its surface. Therefore, the *R*(*Q*(*R*(*Q*(*RW*)))) equivalent electrochemical circuit model is most suitable to fit this measurement, wherein *R*_s_ is the solution resistance in O/W emulsions, and *R*_f_ and *R*_ct_ represent the protective film resistance and charge transfer resistance, respectively. The polarization resistance *R*_po_ (*R*_po_ = *R*_f_ + *R*_ct_) is dominantly controlled by the charge transfer process, as the value of *R*_f_ is relatively smaller. *W* stands for Warburg impedance. *Q*_f_ and *Q*_dl_ are the constant phase elements (CPE), representing the film capacitance and double-layer capacitance, respectively [24]. The parameters of the EIS results of different reduced Cu electrodes in the O/W emulsions are shown in Table 6.

Figure 4b,c presents the Bode absolute plots and Bode phase plots of different rolled Cu electrodes in the O/W emulsions. The impedance values of the Cu samples were found to significantly decrease with cold rolling reduction over the whole frequency range. Meanwhile, their phase angles at low frequencies decreased with the accumulated reduction. Generally, larger values of log |*Z*| always represent a superior protection performance [21], while low phase angle values at low frequencies usually indicate corrosion. From these figures, it can be inferred that the rolling procedure could weaken the corrosion resistance of copper.

### 3.2. Surface Analysis

The Cu electrodes after 2 h of electrochemical measurements were dried subsequently and further observed with a laser scanning confocal microscope. The 3D topographies and height profiles of the Cu samples before and after the electrochemical tests are displayed in Figure 5. As all of the Cu electrodes were fresh polished before the tests, as shown in Figure 5a,c,e,g, their topographies seemed smooth. With the reduction accumulated, a small quantity of shallow rolling trace appeared on the Cu surface. The topographies became much rougher after the electrochemical tests. From the 3D topography in Figure 5b, it is found that some dispersive pits appeared on the raw strip surface. These pits became denser with the increase of the rolling reduction (as shown in Figure 5d,f,h). In particular, while Cu experienced the largest deformation of 77.3% reduction, the small pits aggregated and propagated to form larger corrosion pits. The height profile includes the information of mean height of profile irregularities (*R*_a_), maximum height of profile peak (*R*_p_), and maximum depth of profile valley (*R*_v_) of the Cu samples. As shown in the figures, the *R*_a_ values of uncorroded electrodes were around 0.05 μm. However, with the corrosion occurring, the height profile of these electrodes became fluctuant. It can be seen that each graph exhibits larger absolute values of *R*_v_ than *R*_p_, indicating that pitting corrosion largely exists on the Cu surface. As the rolling reduction increases, the height profile curves of the rolled Cu strips exhibited fluctuations. The topographies became rougher with severer pitting corrosion occurring on the surface.

Figure 6 shows the FE-SEM images of the Cu electrodes under different cold rolling reductions before and after the electrochemical experiments. After mechanical polishing, a small polish trach existed on the electrode surface, and the surface morphologies showed different degrees of pits after the electrochemical tests. The surface morphology of the raw strip electrode (Figure 6b) was relatively uniform and appeared to have less scattered corrosion pits. Corrosion pits exhibited various degrees of expansion as the rolling proceeded. As shown in Figure 6d, 20.7% rolling reduction was found to cause some small pits to connect with other pits. When the reduction increased to 58.6% (Figure 6f), the density of the pits increased significantly, becoming dense and covering the entire surface. These corrosion pits eventually existed in the form of huge pitting corrosion when the Cu strip experienced 77.3% reduction (Figure 6h), and its morphology was seriously damaged under this circumstance.

The corrosion characteristics of these pits were further investigated using high-resolution scanning electron microscopy (HR-SEM). Taking the 58.6% rolled Cu strip as an example, as shown in Figure 7a,b, some large sphere-like pits (marked by white arrows) and small metastable pits (marked by red arrows) were observed on the surface. The diameter of the large pits was around 7 μm, and with metastable pits ranging from 0.7 to 1.3 μm. Clearly, the whole surface could be divided into three parts, including the uncorroded layer, interface, and corroded layer, as shown in Figure 7c. The morphology of the uncorroded layer seemed clean and flat. The interface was loose and accompanied by the occurrence of some cracks. Pits grew on the corroded surface and formed the granulate corrosion products. The EDS surface scanning mappings are shown in Figure 7d, and it can be seen that Cu element was mainly distributed on the uncorroded surface, while the Cl, S, and C elements were mostly distributed on the corroded surface. This phenomenon indicates that pitting corrosion is derived from these Cl, S, and C contents in the O/W emulsions.

The XPS analysis of the corrosion pits on the rolled Cu strip was performed after the morphology observation, and the results are shown in Figure 8. The binding energy of some of the standard compounds of Cu, O, C, S, Cl, and P contents are listed in the plots, which were obtained from the NIST XPS database. Firstly, as shown in Figure 8a, the peaks detected at the binding energy of 952.56 eV, 952.7 eV, and 952.5 eV (Cu 2p1/2) represent metallic Cu, CuO, and Cu_2_O, respectively [26,27,28]. On the other hand, the fitted peak detected at 932.5 eV (Cu 2p3/2) could be constructed by five separation peaks, indicating the presence of some Cu (I) and Cu (II) compounds. In the O 1s spectrum, as Figure 8b shows, the peaks at 532.81 eV were indicative of C = O/C-O organic compounds. The other peak appeared at 532.2 eV, combined with 935.0 eV in the Cu 2p3/2 spectrum (Figure 8a) and 168.15 eV in the S 2p spectrum (Figure 8c), and the existence of CuSO_4_ could be confirmed [29]. Furthermore, the binding energy of the C 1s spectrum (Figure 8d) showed a strong peak at 284.6 eV, representing the existence of C-C and C-H in the emulsifier oils [11]. In addition to the saturated carbon chain, a weak peak was detected at 287.3 eV. This is related to the carbonate groups (${\mathrm{CO}}_{3}^{2-}$), indicating the existence of CuCO_3_ compounds [30]. From Figure 8e, it is seen that P element did not experience any chemical reactions with Cu and the peak detected at 134.1 eV was therefore indicative of undecomposed phosphate [31]. Another important finding is depicted in the Cl 2p spectrum (Figure 8f). The peak was significantly divided into two separation peaks located at 199.1 eV and 198.2 eV, representing CuCl and $\mathrm{Cu}{\mathrm{Cl}}_{2}^{-}$, respectively [32,33]. These compounds were the main factors that caused severe pitting corrosion on the Cu surface.

### 3.3. Microstructure and Mechanical Properties

It is widely believed that cold rolling changes the microstructure and properties of metals. The variation of the metal corrosion property is related to its microstructure [14,34,35]. Figure 9 presents the EBSD mapping for the microstructure of Cu strips under different cold rolling reductions. The statistical data of the grain size and grain boundaries (GBs) variations are shown in Table 7. It is apparent that the average grains size (*D*_adv_) of the Cu strips were refined from 5.21 μm to 2.42 μm, while the total number of GBs increased with the cold rolling operations. As shown in Figure 9a, the grains of the raw Cu strip were relatively homogenous and a small portion of twin boundaries (TBs) occurred at the surface. With the accumulation of the rolling reduction (shown in Figure 9b–d), the grains became blurry and the GBs were gradually broken. Generally, a 15° criterion was employed to define high-angle boundaries (HAGBs) vs. low-angle boundaries (LAGBs) [36]. As shown in Table 5, the fraction of HAGBs was found to decrease with the rolling reduction, meanwhile that of LAGBs increased.

Figure 10 shows the variations of the misorientation distribution and inverse polar figure (IPF) of the rolled Cu strips. The peaks of misorientation distributions were mainly located at 55–65° on the misorientation axis and 0–10° on the misorientation axis. The peak of distribution located at 55–65° on the misorientation axis probably represents the <1 1 1> TBs. It was found that the fraction of 55–65° on the misorientation axis exhibited a reduction tendency (from 39.7 to 5.1%). Similarly, from the IPF results, the maximum values of TBs also reduced from 5.59 to 3.48. As cold deformation promoted the grain boundaries to be elongated or rotated along the rolling direction [37], the role of deformation compatibility caused an increase of LAGBs.

From the perspective of the microstructure, the variations of the grain size and grain boundary were derived from the presence of rolling residual stress on metal, which is the key characteristic that deteriorates the corrosion properties of the material [38,39]. In the case of the rolled Cu strips, the variation of the mechanical properties caused by deformation strengthening played a vital role in its subsequent corrosion.

Table 8 presents the mechanical properties of the near-pit regions on the rolled Cu strip surfaces. The fracture toughness (*K*_IC_) of the Cu strips was found to be decreased with the rolling extensions, while the absolute value of the residual stress (*σ*_γ_) increased. *σ*_γ_ of the 77.3% rolled strip increased to −593.38 MPa, and a negative value represented the compressive stress. In the case of a large reduction, more crystal defects and cracks were be generated at the surface microstructure of the Cu strip. Due to the accumulation of residual stress, it was easy to induce stress corrosion in the material interior, which further increased the corrosive tendency and caused the expansion of the pitting corrosion [40].

### 3.4. FIB and TEM Results

In order to give an in-depth study on the corrosion mechanism of Cu strips under different cold rolling reductions, the FIB technique was performed to characterize the cross-sectional microstructure of the corrosion pits. The cross-section microstructure of the corrosion pits on the 20.7% reduced Cu sample is shown in Figure 11a. It was found that the size along the depth of the entire pit was around 2.32 μm. A small portion of metastable pits and microcracks was observed inside the corrosion pit. Nevertheless, as the cold rolling reduction accumulated to 77.3% (Figure 11b), the depth of pit increased to 3.18 μm. Meanwhile, the metastable pits and microcracks became more intensive and larger accordingly. These metastable pits and microcracks were distributed along the rolling directions, indicating that pitting corrosion tends to be distributed in regions with more defects and deformations.

A similar phenomenon can be observed from the TEM micrographs. The dislocation configurations on the microstructure of the 20.7% reduced Cu strip are shown in Figure 12a,b. It can be seen that only a few dislocation sources appeared in the uncorroded site, whereas a large number of dislocation cells appeared in the near-pits region. As for the 77.3% reduced strip (shown in Figure 12c), some modulated structures were observed in the uncorroded region, which could be regarded as the sub-structure that appeared at the local region. This phenomenon is related to the broken of grain boundaries caused by large plastic deformation. Additionally, in the near-pits region (Figure 12d), the number of dislocations was found to increase remarkably in comparison with the 20.7% reduced strip. It can be clearly seen that a long dislocation wall formed on the microstructure by dislocation entanglements. These results are consistent with the previous surface and microstructure observations.

### 3.5. Discussions

On the basis of the above results, it was found that the corrosivity tendency of Cu strips under different cold rolling reductions (ε) in O/W emulsions followed the order of ε_0%_ < ε_20.7%_ <ε_50.6%_ < ε_77.3%_., The residual stress on the surface microstructure increased due to the cold rolling deformations, resulting in the refinement of grains, broken of GBs, increase of LAGBs, and decrease of HAGBs. Furthermore, surface defects such as metastable pits, microcracks, and dislocation cells were found to increase with the reduction accumulations. With the role of O/W emulsions, Cl, C, and S elements were significantly distributed on the corroded surface. Among them, Cu was likely to react with the ${\mathrm{Cl}}_{}^{-}$ component in the anodic region [41,42]:(5)$$\mathrm{Cu}+{\mathrm{Cl}}^{-}\to \mathrm{Cu}\mathrm{Cl}+e^{-}$$
(6)$$\mathrm{Cu}\mathrm{Cl}+{\mathrm{Cl}}^{-}\to \mathrm{Cu}{\mathrm{Cl}}_{2}^{-}$$
(7)$$\mathrm{Cu}{\mathrm{Cl}}_{2}^{-}\to {\mathrm{Cu}}^{2+}+2{\mathrm{Cl}}^{-}+e^{-}$$

Meanwhile, the oxygen in the emulsion solutions as well as other anions such as ${\mathrm{SO}}_{4}^{2-}$ and ${\mathrm{CO}}_{3}^{2-}$ also play significant roles in the electrode reaction:(8)$$2\mathrm{Cu}+1/2O_{2}\to {\mathrm{Cu}}_{2}O$$
(9)$$2\mathrm{Cu}{\mathrm{Cl}}_{2}^{-}+H_{2}O\to {\mathrm{Cu}}_{2}O+2H^{+}+4{\mathrm{Cl}}^{-}$$
(10)$${\mathrm{Cu}}^{2+}+2{\mathrm{RCOO}}^{-}\to \mathrm{Cu}{\mathrm{CO}}_{3}+R–R+\mathrm{CO}$$
(11)$${\mathrm{Cu}}^{2+}+{\mathrm{SO}}_{4}{}^{2-}\to \mathrm{Cu}{\mathrm{SO}}_{4}$$

Furthermore, the corrosion products were proven to be CuCl, $\mathrm{Cu}{\mathrm{Cl}}_{2}^{-}$, Cu_2_O, CuCO_3_, and CuSO_4_ [11,30,43,44,45], appearing in the form of pitting corrosion on the Cu surface. These corrosion products eventually caused the presence of large cathodic areas surrounded by small anodic corrosion sites [46]. Therefore, under the interactive effect of pitting corrosion and stress corrosion, pits expanded along the rolling direction. Anions aggregated in surface defects such as dislocations, metastable pits, and microcracks, thus the corrosion eventually evolved into pitting corrosion, which is more serious. The mode of the whole corrosion behavior of the Cu strip rolled with O/W emulsions is depicted as Figure 13a, and the mechanism of corrosion expansion is shown in Figure 13b.

## 4. Conclusions

In this paper, the effect of cold rolling reduction on the pitting corrosion behavior and microstructure of Cu strips in O/W emulsions were systematically investigated through a combination of electrochemical experiments, surface and corroded microstructural observations, and mechanical property analysis. The primary conclusions are drawn as follows:(1)The electrochemical results show the corrosion current densities of Cu strips in the O/W emulsions increased with accumulated reduction, while the corrosion potentials shifted towards being more negative in the anodic region. The pitting potentials and corrosion resistances were both decreased with the processing of cold rolling. These phenomena demonstrated that the corrosive tendency of Cu strips under different rolling reduction (ε) followed the order of ε_0%_ < ε_20.7%_ < ε_50.6%_ < ε_77.3%_.(2)Surface observations indicated that there were different degrees of pit expansions that occurred on the rolled Cu surfaces. These pits became denser and the surface became rougher with the increase of rolling reduction. Some metastable pits existed in the larger pits. Cu reacted easily with chlorine, sulfur, and carbon components from the O/W emulsions, Then, the corrosion products, i.e., CuCl, $\mathrm{Cu}{\mathrm{Cl}}_{2}^{-}$, Cu_2_O, CuCO_3_, and CuSO_4_ were generated, which appeared in the form of pitting corrosion on the Cu surface.(3)EBSD mappings demonstrated the average grain size of Cu strips refined from 5.21 μm to 2.42 μm with the accumulated reduction. The fraction of low-angle boundaries increased, while that of high-angle boundaries decreased. This was due to the accumulation of residual stress, which induced stress corrosion in the material interior, further increased the corrosive tendency, and contributed to the expansion of the pitting corrosion.(4)From the TEM and FIB characterizations, it is concluded that corrosion pits expanded along the rolling direction under the interactive effect of pitting corrosion and stress corrosion. Meanwhile, anions aggregated in the surface defects, such as dislocations, metastable pits, and microcracks, which thereby accelerated the pitting corrosion of the surface.

## Figures and Tables

**Figure 1 materials-14-07911-f001:**
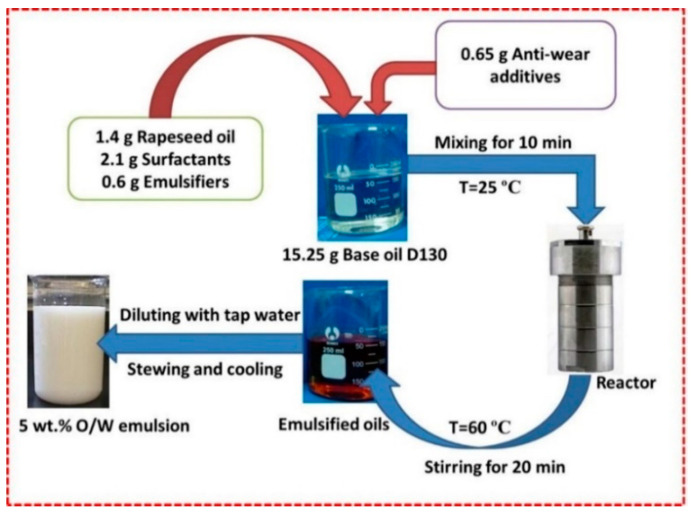
Preparation process of O/W emulsion.

**Figure 2 materials-14-07911-f002:**
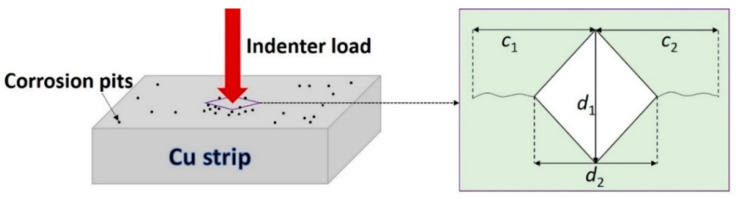
Principle of the Vickers indentation methods.

**Figure 3 materials-14-07911-f003:**
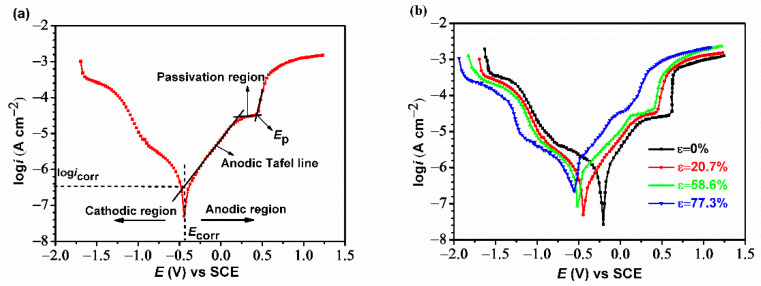
Potentiodynamic polarization results of Cu electrodes in the O/W emulsions: (**a**) Tafel extrapolation method of the anodic polarization curve and (**b**) polarization curves under different rolling reductions.

**Figure 4 materials-14-07911-f004:**
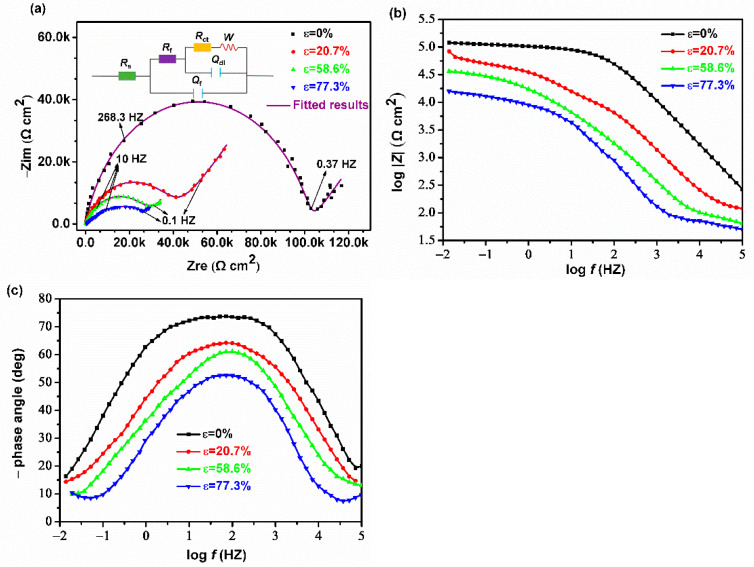
EIS results of Cu electrodes under different rolling reductions in the O/W emulsions: (**a**) Nyquist plot, (**b**) Bode absolute plots, and (**c**) Bode phase plots.

**Figure 5 materials-14-07911-f005:**
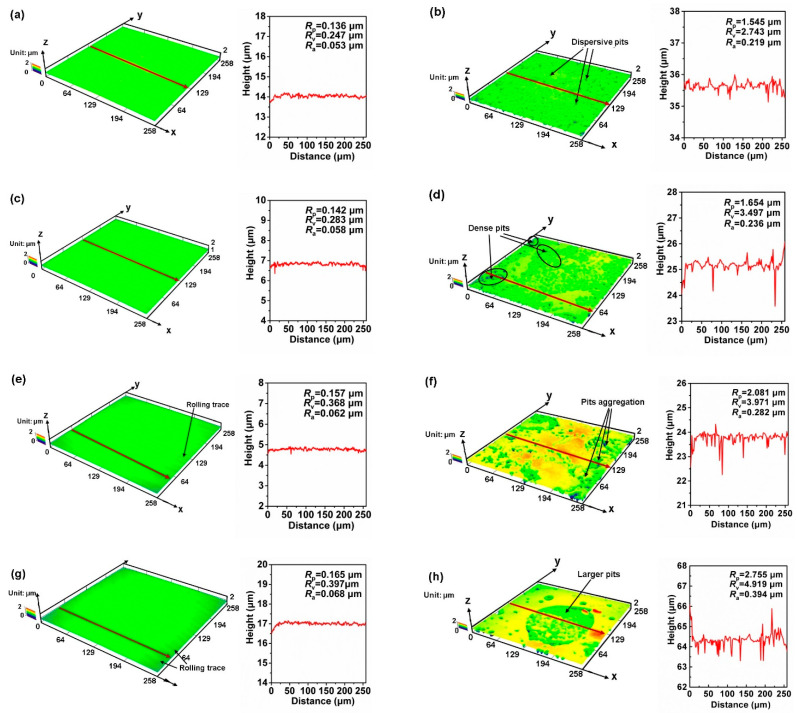
The 3D topographies and height profiles of the Cu electrodes before and after the electrochemical tests under different rolling reductions: (**a**,**b**) 0%, (**c**,**d**) 22.7%, (**e**,**f**) 58.6%, and (**g**,**h**) 77.3%.

**Figure 6 materials-14-07911-f006:**
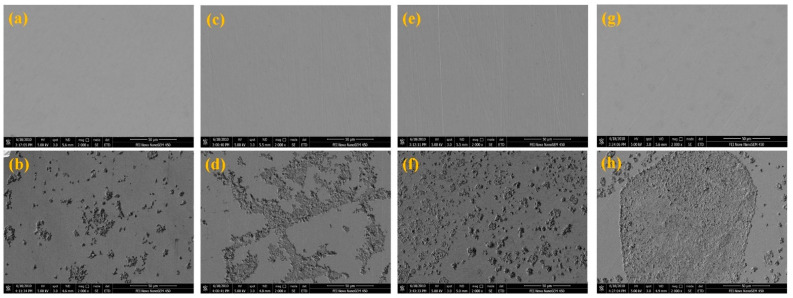
SEM morphologies of Cu electrodes before and after the electrochemical tests under different rolling reductions: (**a**,**b**) 0%, (**c**,**d**) 22.7%, (**e**,**f**) 58.6%, and (**g**,**h**) 77.3%.

**Figure 7 materials-14-07911-f007:**
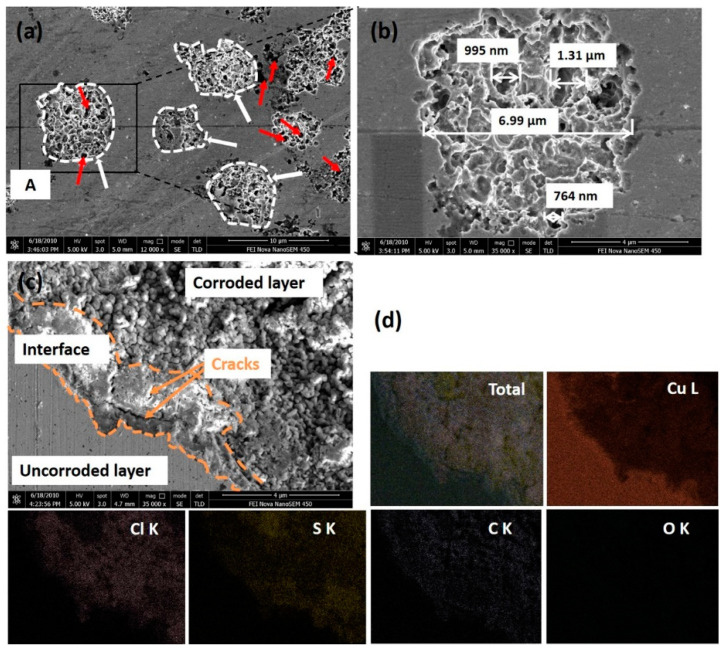
HR-SEM images and EDS analysis of the 58.6% reduced Cu surface after the electrochemical experiments: (**a**) HR-SEM images of the corrosion pits; (**b**) magnification of site A in Figure 7a; (**c**) morphology of corroded layer, interface and uncorroded layer; and (**d**) EDS surface scanning mappings of Figure 7c.

**Figure 8 materials-14-07911-f008:**
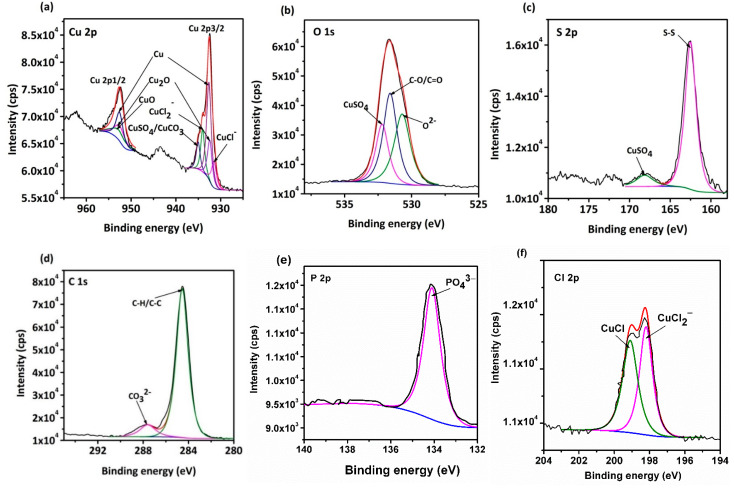
XPS spectra of the corrosion pits of the Cu strip after the electrochemical experiment: (**a**) Cu 2p, (**b**) O 1s, (**c**) S 2p, (**d**) C 1s, (**e**) P 2p, and (**f**) Cl 1s.

**Figure 9 materials-14-07911-f009:**
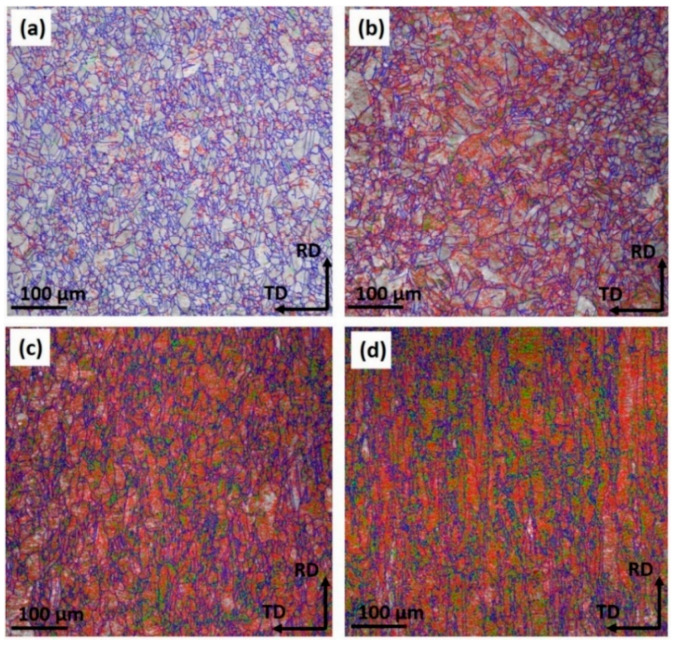
EBSD mapping of microstructures for Cu strips under different cold rolling reductions after being corroded by O/W emulsions. (**a**) 0% reduction, (**b**) 22.7% reduction, (**c**) 58.6% reduction, and (**d**) 77.3% reduction (in the EBSD mappings, red, green, and blue lines represent the angle of 2–5°, 5–15°, and 15–180° boundaries, respectively.).

**Figure 10 materials-14-07911-f010:**
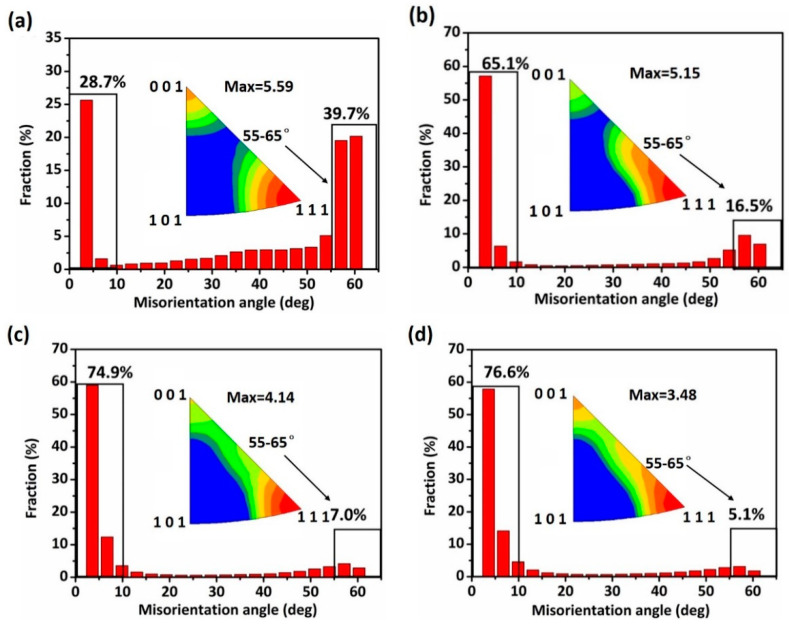
Misorientation distribution and inverse polar figures of Cu strips under different cold rolling reductions: (**a**) 0% reduction, (**b**) 22.7% reduction, (**c**) 58.6% reduction, and (**d**) 77.3% reduction.

**Figure 11 materials-14-07911-f011:**
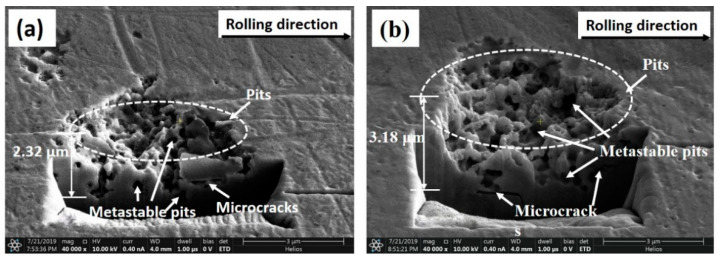
FIB-SEM cross-section microstructure of the corrosion pits of Cu strips under small and large reductions: (**a**) 22.7% reduction and (**b**) 77.3% reduction.

**Figure 12 materials-14-07911-f012:**
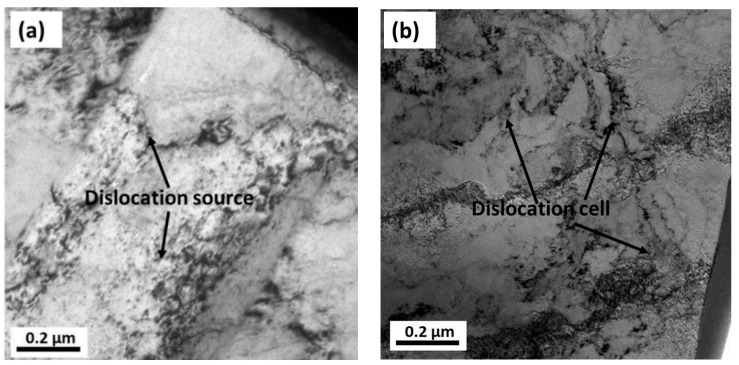
TEM bright field images of different regions of the rolled Cu strips: (**a**) uncorroded site of 20.7% reduced strip, (**b**) near-pit site of 20.7% reduced strip, and (**c**) uncorroded site of the 77.3% reduced strip, (**d**) near-pit site of 77.3% reduced strip.

**Figure 13 materials-14-07911-f013:**
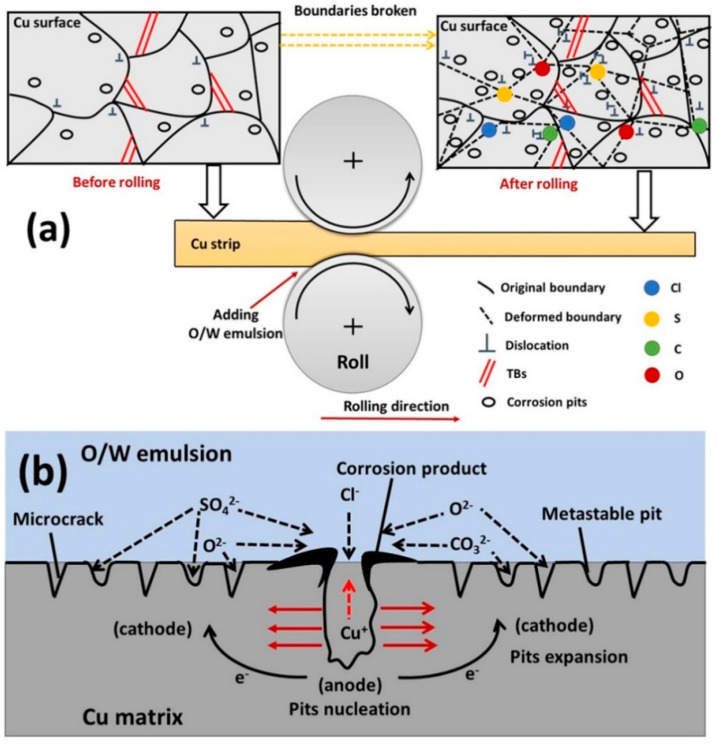
(**a**) Mode of the whole corrosion behavior of a Cu strip rolled with O/W emulsions; (**b**) mechanism of pitting corrosion expansion.

**Table 1 materials-14-07911-t001:** Component of the raw Cu strip in this investigation.

**Component**	Cu	Pb	S	Cd	P	Fe
**Wt.%**	99.9900	0.0003	0.0016	0.0002	0.0013	0.0009

**Table 2 materials-14-07911-t002:** Details on composition and physiochemical properties of mineral oil D130 and rapeseed oil.

Properties	Mineral Oil D130	Rapeseed Oil
Main compositions	Direct alkane, Branched alkane, Cycloalkanes	Erucic acid, Arachidic acid, Linoleic acid, linolenic acid
Aromatics content (wt.%)	0.5	-
Sulfur content (wt.%)	<0.1	<0.1
Phosphorous content (wt.%)	<0.1	<0.1
Viscosity for 40 °C (mm^2^/s)	6.12	13.5~14.0
Flash point (°C)	>140	>110
Suppliers	Sinopec Group Shanghai Co., Ltd., Shanghai, China	Red Oil Chengdu Ltd., Chengdu, China
Year of production	2017	2019

**Table 3 materials-14-07911-t003:** Ionic concentration of the prepared O/W emulsion.

**Ionic Types**	${\mathrm{Cl}}^{-}$	${\mathrm{NO}}_{3}^{-}$	${\mathrm{SO}}_{4}^{2-}$	${\mathrm{PO}}_{4}^{3-}$	$K^{+}$	${\mathrm{Ca}}^{2+}$	${\mathrm{Na}}^{+}$	${\mathrm{Mg}}^{2+}$
**Concentrations (ppm)**	15.62	21.78	71.52	6.35	-	67.26	13.94	6.39

**Table 4 materials-14-07911-t004:** Actual rolled thickness of each pass.

**Rolling Pass**	0	1	2	3	4	5
**Thickness (mm)**	1.98	1.85	1.57	1.16	0.82	0.45

**Table 5 materials-14-07911-t005:** Potentiodynamic polarization parameters of Cu electrodes under different rolling reduction.

Reduction	*E*_corr_ (mV)	*i*_corr_ (A/cm^2^)	*β*_a_ (V/dec)	*β*_c_ (V/dec)	*E*_p_ (V)
0%	−134	1.23 × 10^−6^	3.09	−1.92	0.58
20.7%	−425	9.55 × 10^−6^	2.98	−2.01	0.45
58.6%	−505	1.76 × 10^−5^	2.75	−2.09	0.41
77.3%	−527	2.56 × 10^−5^	2.99	−1.98	0.11

**Table 6 materials-14-07911-t006:** Parameters of the EIS results of different reduced Cu electrodes in O/W emulsions.

Reduction	*R*_ct_ (kΩ cm^2^)	*R*_f_ (kΩ cm^2^)	*R*_po_ (kΩ cm^2^)	*R*_s_ (kΩ cm^2^)	*Q*_f_ (μF cm^−2^)	*Q*_dl_ (μF cm^−2^)	*W* (×10^−2^ Ω cm^2^ S^1^^/2^)
0%	107.82	1.12	108.94	2.81	12.68	25.64	18.62
20.7%	44.08	0.42	44.50	2.82	18.86	38.40	15.64
58.6%	28.31	0.38	28.69	2.81	24.42	50.42	31.08
77.3%	20.27	0.32	20.59	2.82	28.82	340.64	40.12

**Table 7 materials-14-07911-t007:** Statistical data of the grain size and grain boundary variations.

**Reduction**	0%	20.7%	58.6%	77.3%
***D*_adv_ (μm)**	5.21	5.14	3.08	2.42
**Number of GBs**	285,646	463,615	971,524	13,543,64
**Number of LAGBs**	82,266	305,522	745,159	10,672,39
**Fraction of LAGBs (%)**	28.8	65.9	76.7	78.8
**Number of HAGBs**	203,380	158,093	226,365	287,125
**Fraction of HAGBs (%)**	71.2	34.1	23.3	21.2

**Table 8 materials-14-07911-t008:** Mechanical properties of the near-pit regions on Cu strip surfaces with different reductions.

ε	*D*/μm	*H*/MPa	*C*/μm	*K*_IC_/MPa·m^1/2^	*σ*_γ_/MPa
0%	165.79	221.76	69.48	15.38	−35.69
20.7%	1297.67	251.64	106.76	11.25	−238.43
58.6%	356.36	286.67	139.47	8.78	−296.45
77.3%	386.37	357.73	156.98	3.21	−598.38

## Data Availability

The data presented in this work are available on request from the corresponding author.

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
