# Peer review of "Pitting Corrosion Behavior and Surface Microstructure of Copper Strips When Rolled with Oil-in-Water Emulsions"

_materials, 2021, doi:10.3390/ma14247911_

Round 1
Reviewer 1 Report
I have read the manuscript titled ''pitting corrosion behavior and surface microstructure of copper strips when rolled with oil-in-water emulsions''. The manuscript is well written with minor correction.
Please remove etc, line 64. etc could mean anything
The position of your ion in line 175 is confusing, and also may be you will need to refer to equation 6 for better understanding
Page 7, line 226-227, From the 3D topography in Figure 5 (a), it is found that there are some dispersive pits appear on the raw strip surface. May be you need to depict these pits with arrows or something. It would be dodgy for non experts to see these pits you are referring to.
Line 289, carbonates group, remove s and change to carbonate groups
Line 295 ion, CuCl2-, the position of your ion is confusing.
Author Response
Response to Reviewer 1 comments
Comments and Suggestions for Authors: I have read the manuscript titled ''pitting corrosion behavior and surface microstructure of copper strips when rolled with oil-in-water emulsions''. The manuscript is well written with minor correction.
Point 1: Please remove etc, line 64. etc could mean anything.
Response to point 1: We have removed “etc” in line 30 and line 64 in our revised version according to your suggestion.
Point 2: The position of your ion in line 175 is confusing, and also maybe you will need to refer to equation 6 for better understanding.
Line 295 ion, CuCl2- the position of your ion is confusing.
Response to point 2: As equation 6 explained, derives from the reaction involving CuCl transformation into a cuprous complex .The process is the second chemical reaction in solutions that contained copper and chloride ion. These phenomena are also proved by many other researchers [20, 21]. It may be a confusing caused by writing mistakes. Therefore, we have corrected all of the spelling and writing mistakes on the ions in the entire paper to avoid the misunderstanding of the readers.
Relevant reference:
[20] Electrochim. Acta 1998, 43, 2469–2483.
[21] RSC Adv. 2015, 5, 63866–63873.
Point 3: Page 7, line 226-227, From the 3D topography in Figure 5 (a), it is found that there are some dispersive pits appear on the raw strip surface. Maybe you need to depict these pits with arrows or something. It would be dodgy for non experts to see these pits you are referring to.
Response to point 3: According to your kind suggestion, we have inserted black arrows for depicting these dispersive pits, as shown in the revised version of this manuscript.
Point 4: Line 289, carbonates group, remove s and change to carbonate groups.
Response to point 4: We have revised in “carbonate groups” in line 314.
In addition, we also have spelling checked and polished the English.
The revised version of the manuscript is seen in the attachment.

Reviewer 2 Report
I suggest that a cyclic polarization be made in order to be able to
see the values of the potentials at which the pit occurs and to determine
the protection potential.
Author Response
Response to Reviewer 2 comments
Point1: I suggest that a cyclic polarization be made in order to be able to see the values of the potentials at which the pit occurs and to determine the protection potential.
Response to point 1: Thanks for your suggestion. Cyclic polarization is indeed one of the reasonable appropriate methods to reflect the pitting potentials. In this paper, we use potentiodynamic polarization curves, which can not only see the pitting potentials, but also include the influence of other useful parameters, such as corrosion potential (Ecorr), corrosion current density (icorr), anodic Tafel slope (βa) and cathodic Tafel slope (βc), etc. These parameters also contribute to the corrosion behavior of the Cu samples in O/W emulsion solutions, especially the anodic and cathodic polarization phenomena. Thus, potentiodynamic polarization maybe more appropriate in this study. There are also other studies using potentiodynamic polarization method to deal with pitting corrosion/corrosion inhibition properties of metals in solutions.
Relevant reference:
Corros. Sci. 2015, 94, 156–164.
Corros. Sci. 2016, 112, 233–240.
Electrochim. Acta 2016, 191, 247–255.
According to your advice, we mark the values of pitting potentials (Ep) into Table 3 to determine the protection potential. And we will also consider to carry out cyclic polarization method in our further research as you mentioned.
The revised version of the manuscript is seen in the attachment.

Reviewer 3 Report
The manuscript is a consistently conducted and documented scientific experiment. The authors used many advanced research tools, including electrochemical polarization methods. With reference to the results of the impedance tests presented in Fig. 4, some questions arise.
1. Which electrochemical parameter in the equivalent electrical system is used to identify the diameter of the flattened semicircle of the spectrum in the Nyquist plot (Fig. 4a)? This information should be included in the text of the manuscript.
2. In the Bode diagram (Fig. 4c) in the low-frequency range at frequencies of the order of 0.01 Hz, low phase angle values usually indicate corrosion. I would like your comment regarding the obtained impedance measurement results.
Author Response
Response to Reviewer 3 comments
Comments: The manuscript is a consistently conducted and documented scientific experiment. The authors used many advanced research tools, including electrochemical polarization methods. With reference to the results of the impedance tests presented in Fig. 4, some questions arise.
Point 1: Which electrochemical parameter in the equivalent electrical system is used to identify the diameter of the flattened semicircle of the spectrum in the Nyquist plot (Fig. 4a)? This information should be included in the text of the manuscript.
Response to point 1: As our former statements in 3.1 in the manuscript, “Generally, the semicircles are related to charge transfer resistance and double layer capacitance”, which is also approved by other references [23-25]. Therefore, Rct, Qdl (Rct represents charge transfer resistance, Qdl is the constant phase element, reflecting double layer capacitance) are commonly used to identified diameter of the flattened semicircle of the spectrum in the Nyquist plot. We supplemented “Table 6 EIS parameters of different reduced Cu electrodes in O/W emulsions” and added references to enrich our findings.
Relevant reference:
[23] Corros. Sci. 2017, 126, 295–304.
[24] Corros. Sci. 2014, 85, 77–86.
[25] Corros. Sci. 2017, 119, 68–78.
Point 2: In the Bode diagram (Fig. 4c) in the low-frequency range at frequencies of the order of 0.01 Hz, low phase angle values usually indicate corrosion. I would like your comment regarding the obtained impedance measurement results.
Response to point 2: we supplemented the comments “Meanwhile, their phase angles at low frequencies decrease with accumulated reduction. Generally, the larger values of log |Z| always represent superior protection performance, while low phase angle values at the low frequencies usually indicate corrosion.” in the revised version.
The revised version of the manuscript is seen in the attachment.

Reviewer 4 Report
Reviewer Recommendation and Comments for manuscript materials-1494094 with the title: “Pitting corrosion behavior and surface microstructure of copper strips when rolled with oil-in-water emulsions”, authors: Xudong Yan and Jianlin Sun.
The authors present the corrosion processes of copper strips when are rolled with oil/water (O/W) emulsion lubricant. Cu strips are subjected to rolling tests and an O/W emulsion is used as lubricant. Electrochemical measurements were performed to determine the electrochemical parameters of Cu in O/W as corrosion environment. Surface morphology of Cu plates after corrosion was analyzed using SEM/XPS methods. Cu plates were subjected to EBSD and mechanical tests.
The article may be published after REVISION.
The main comments that I find useful for improving the quality of the article are presented below:
#2.1. Materials and O/W emulsions preparation
Two important and unknown aspects are the compositions of mineral oil and that of rapeseed oil. It is very important to provide as many details as possible about compositions, manufacturers, suppliers, year of production, etc.
#2.3. Electrochemical measurement
It is not clear whether OCP, EIS and LP were performed on the same Cu sample and in the same solution. All three methods were performed on the same Cu plate? in the same solution? New copper samples were used for each electrochemical analysis? Fresh solutions were used every time?
#Figure 3. (insert of Figure 3a)
icorr must be replaced with logicorr.
#line 184: „With the accumulation of rolling reduction, the values of icorr gradually increase and the Ecorr shift towards more negative.”
There is a contradiction between the polarization curves shown in Figure 3b and the values of the corrosion current density. According to Figure 3b, especially for intermediate samples (reduction of 20.7 and 58.6%), the polarization curves are almost overlapping. This figure highlights the change in both the anodic and the cathodic process. The extent to which the anodic process is activated is the same with the extent to which the cathodic process is reduced, so that no significant changes in the corrosion rate can be observed. The increase in the corrosion rate especially for the sample of 77.3% is not in line with the results obtained by the EIS measurements. It is recommended to introduce a table with EIS parameters of curves from Figure 4a, b, c.
#3.2 Surface analysis
It is recommended to also introduce the topographies of the Cu surfaces before the electrochemical test (a-0% reduction – before and after the electrochemical test, b-22.7% reduction - before and after the electrochemical tests, c-.....) and the appropriate comments.
#SEM analysis
It is recommended to also introduce the topographies of the Cu surfaces before the electrochemical test (a-0% reduction – before and after the electrochemical test, b-22.7% reduction - before and after the electrochemical tests, c-.....) and the appropriate comments.
#XPS analysis (lines 290-292) “…is therefore indicative of undecomposed phosphate…” – Figure 8e.
(line 96) Table 2. Ionic concentration of the prepared O/W emulsion.
The reader may be confused; Table 2 does not indicate the presence of the phosphate ion, while XPS analysis indicates the presence of this ion. The results of the analyzes should present the same conclusions.
# 3. Results and discussion and 3.5. Discussions?
#line 390
Reaction (10) must be verified!
#The typos must be corrected.
…anions ions…
etc.
#There are some grammar and typing mistakes.
#The authors must revise the entire manuscript.
Author Response
Response to Reviewer 4 comments
Comments:the authors present the corrosion processes of copper strips when are rolled with oil/water (O/W) emulsion lubricant. Cu strips are subjected to rolling tests and an O/W emulsion is used as lubricant. Electrochemical measurements were performed to determine the electrochemical parameters of Cu in O/W as corrosion environment. Surface morphology of Cu plates after corrosion was analyzed using SEM/XPS methods. Cu plates were subjected to EBSD and mechanical tests.
The article may be published after REVISION.
The main comments that I find useful for improving the quality of the article are presented below:
Point 1: 2.1. Materials and O/W emulsions preparation
Two important and unknown aspects are the compositions of mineral oil and that of rapeseed oil. It is very important to provide as many details as possible about compositions, manufacturers, suppliers, year of production, etc.
Response to point 1: We have supplemented Table 2 “Details in composition and physiochemical properties of mineral oil D130 and rapeseed oil” to give the important information of these two oils. In addition, we revised the statements in section 2.1 “Materials and O/W emulsions preparation”. Such information of compositions, manufacturers or suppliers, year of production of other reagents were also included in the revised version.
Table 2. Details in composition and physiochemical properties of mineral oil D130 and rapeseed oil.
|
Properties |
Mineral oil D130 |
Rapeseed oil |
|
Main compositions |
Direct alkane; Branched alkane; Cycloalkanes |
Erucic acid; Arachidic acid; Linoleic acid; Linolenic acid |
|
Aromatics content (wt.%) |
0.5 |
- |
|
Sulfur content (wt.%) |
<0.1 |
<0.1 |
|
Phosphorous content (wt.%) |
<0.1 |
<0.1 |
|
Viscosity for 40 °C( mm2/s) |
6.12 |
13.5~14.0 |
|
Flash point(°C) |
>140 |
>110 |
|
Suppliers |
Sinopec Group Shanghai Co., Ltd. |
Red Oil Chengdu Ltd |
|
Year of production |
2017 |
2019 |
Point 2: #2.3. Electrochemical measurement
It is not clear whether OCP, EIS and LP were performed on the same Cu sample and in the same solution. All three methods were performed on the same Cu plate? in the same solution? New copper samples were used for each electrochemical analysis? Fresh solutions were used every time?
Response to point 2:
In this work, the rolled Cu strips under different reductions (0%, 20.7%, 58.6%, 77.3%) were used as 4 samples. They were polished for the same roughness and prepared as electrodes, respectively. For each sample, (e.g. 0% reduced Cu strip → electrode→OCP test→EIS test→LP test→ 3D topography observation→SEM observation; 20.7% reduced Cu strip → electrode→OCP test→EIS test→LP test→ 3D topography observation→SEM observation…..) We also have prepared sufficient O/W emulsions and divided them into 4 parts, each of the Cu electrode was immersed in the same emulsion solution for OCP, EIS and other experiments.
We carried out the experiments in strict accordance with the control variable method, the “rolling reduction” is the only variable. This can efficiently avoid other variables’ influences.
In the revised version, we have supplemented the experimental methods, highlighted the details of the preparation of samples to avoid misunderstandings of readers.
Point 3: #Figure 3. (insert of Figure 3a)
icorr must be replaced with logicorr.
#line 184: “With the accumulation of rolling reduction, the values of icorr gradually increase and the Ecorr shift towards more negative.”
There is a contradiction between the polarization curves shown in Figure 3b and the values of the corrosion current density. According to Figure 3b, especially for intermediate samples (reduction of 20.7 and 58.6%), the polarization curves are almost overlapping. This figure highlights the change in both the anodic and the cathodic process. The extent to which the anodic process is activated is the same with the extent to which the cathodic process is reduced, so that no significant changes in the corrosion rate can be observed. The increase in the corrosion rate especially for the sample of 77.3% is not in line with the results obtained by the EIS measurements. It is recommended to introduce a table with EIS parameters of curves from Figure 4a, b, c.
Response to point 3:
We have replaced the icorr with “logicorr” in Figure 3a.
We have revised the comments as “With the accumulation of rolling reduction, logicorr gradually increase and the Ecorr shift towards more negative in the anodic region” in the revised version of this manuscript.
Although there is part of overlapping on the polarization curves, the variation on the anodic reaction of these four reduced electrodes were slightly obvious than the cathodic reaction. We mainly focus on the anodic reaction. It is clearly to be seen the tendency on the values of logicorr follows the order of black<red <green<blue curve in the anodic region. Thus, with the accumulated reduction, the logicorr of Cu electrode increases, the icorr increases, and the Ecorr decreases. In Fig. 4, the corrosion resistance of 77.3% Cu samples is the lowest, reflecting that the fast corrosion rate of this sample, which is in accordion with the former polarization results.
We greatly appreciate your suggestions, and we newly added “Table 6 Parameters of the EIS results of different reduced Cu electrodes in O/W emulsions” to enrich our findings.
Table 6. Parameters of the EIS results of different reduced Cu electrodes in O/W emulsions
|
Reduction |
Rct (kΩ cm2) |
Rf (kΩ cm2) |
Rpo (kΩ cm2) |
Rs (kΩ cm2) |
Qf (μF cm-2) |
Qdl (μF cm-2) |
W (×10-2 Ω cm2 S1/2) |
|
0% |
107.46 |
1.12 |
108.58 |
2.81 |
12.68 |
25.64 |
18.62 |
|
20.7% |
40.88 |
0.42 |
41.3 |
2.82 |
18.86 |
38.40 |
15.64 |
|
58.6% |
28.34 |
0.38 |
28.72 |
2.81 |
24.42 |
50.42 |
31.08 |
|
77.3% |
20.38 |
0.32 |
20.7 |
2.82 |
28.82 |
340.64 |
40.12 |
Point 4: #3.2 Surface analysis
It is recommended to also introduce the topographies of the Cu surfaces before the electrochemical test (a-0% reduction – before and after the electrochemical test, b-22.7% reduction - before and after the electrochemical tests, c-.....) and the appropriate comments.
Response to point 4: We have supplemented the 3D topographies and surface roughness data of the 4 samples before the electrochemical test, some contents are revised. “The 3D topographies and height profile of the Cu samples before and after the electrochemical tests are displayed in Figure 5. Since all Cu electrodes were fresh polished before the tests, as shown in Figure 5(a), (c), (e) and (g), their topographies seem smooth. With the reduction accumulated, part of shallow rolling trace appears on the Cu surface. The topographies become rough after the electrochemical tests.”“As shown in the figures, the Ra of uncorroded electrodes are around 0.05 μm. However, with the corrosion occurring, the height profile of these electrodes become fluctuant.”
Figure 5 The 3D topographies and height profile of Cu electrodes before and after the electrochemical tests under different rolling reduction: (a)~(b) 0%; (c) ~(d)22.7%; (e)~(f) 58.6%; (g)~(h) 77.3%.
#SEM analysis
Point 5: It is recommended to also introduce the topographies of the Cu surfaces before the electrochemical test (a-0% reduction – before and after the electrochemical test, b-22.7% reduction - before and after the electrochemical tests, c-.....) and the appropriate comments.
Response to point 5: We have supplemented the SEM microscopes of the 4 samples before the electrochemical test, some contents are revised. “Figure 6 shows the FE-SEM images of Cu electrodes under different cold rolling reduction before and after the electrochemical experiments. After mechanical polishing, there small polish trach exists on the electrode surface. And the surface morphologies show different degrees of pits after the electrochemical tests.”
This article mainly investigated the effect of rolling reduction on the microstructure (e.g. grain size, grain boundaries, defects, mechanical properties) and corrosion behavior on the Cu strip in O/W emulsions. In order to control variables, the four electrode samples were polished to ensure consistent surface roughness before the tests. Therefore, the surface morphologies of the fresh polished electrodes (electrodes before the electrochemical tests) look similar. And we focus on comparing the surface topographies/morphologies of different reduced samples after corrosion.
Figure 6 SEM morphologies of Cu electrodes before and after the electrochemical tests under different rolling reduction: (a)~(b) 0%; (c) ~(d)22.7%; (e)~(f) 58.6%; (g)~(h) 77.3%.
Point6 #XPS analysis (lines 290-292) “…is therefore indicative of undecomposed phosphate…” – Figure 8e.
(line 96) Table 2. Ionic concentration of the prepared O/W emulsion.
The reader may be confused; Table 2 does not indicate the presence of the phosphate ion, while XPS analysis indicates the presence of this ion. The results of the analyzes should present the same conclusions.
Response to point 6: the concentration of phosphate ion in the O/W emulsion was supplemented in Table 3.
Point 7: # 3. Results and discussionand 3.5. Discussions
#line 390
Reaction (10) must be verified!
Response to point 7:We have deleted reaction (10) and cited the following relevant references to verify the other reactions. We also marked these references in the revised version to avoid the misunderstanding of the readers.
Relevant references:
[11] RSC Adv. 2018, 8, 9833–9840.
[30] Corros. Sci. 2013, 67, 50–59.
[43] Corros. Sci. 2012, 61, 53–62.
[44] Corros. Sci. 2002, 44, 2507–2528.
[45] Corros. Sci. 2006, 48, 2867–2881.
Point 8: #The typos must be corrected. anions ions…etc.
Response to point 8:We have corrected the typos mistakes and revised as “anions”
Point 9:
#There are some grammar and typing mistakes.
#The authors must revise the entire manuscript.
Response to point 9: We are very sorry for our grammar and typing mistakes, we have examined the entire manuscript, and revised these mistakes. We also send this manuscript to native English-speaking colleague to polish the English.
The revised version of the manuscript and details of response are seen in the attachments.

Round 2
Reviewer 4 Report
The authors performed the requested revisions. I think the article is much improved.
However, the authors still have to make a small revision.
The experimental results must lead to the same conclusion.
Please match the Rct values from Table 6 with the EIS results from Figure 4a and with the icorr values from Table 5.
Author Response
Comments:The authors performed the requested revisions. I think the article is much improved. However, the authors still have to make a small revision.
Point 1: The experimental results must lead to the same conclusion.
Response to Point 1: We revised and enriched the conclusions to made it consistent with the previous experimental results.
- Conclusions
In this paper, the effect of cold rolling reduction on the pitting corrosion behavior and corroded microstructure of Cu strips in O/W emulsions were systematically investigated by the combination of electrochemical experiments, surface and corroded microstructural observations and mechanical properties analysis. The primary conclusions are drawn as follows:
- The electrochemical results show the corrosion current densities of Cu strips in the O/W emulsions increased with accumulated reduction, while the corrosion potentials shifted towards more negative in the anodic region. The pitting potentials and corrosion resistances were both decreased with processing of cold rolling. These phenomena demonstrated that the corrosive tendency of Cu strips under different rolling reduction (ε) followed the order of ε0% <ε7% <ε50.6% <ε77.3%.
- Surface observations indicated that there were different degrees of pits expansions occurred on the rolled Cu surfaces. These pits became denser and the surface became rougher with the increase of rolling reduction. Some metastable pits existed in the larger pits. Cu reacted easily with chlorine, sulfur and carbon components from O/W emulsions. Then, the corrosion products including CuCl, CuCl2-, Cu2O, CuCO3 and CuSO4 were generated, which appeared in the form of pitting corrosion on Cu surface.
- EBSD mappings demonstrated the average grain size of Cu strips refined from 5.21 μm to 2.42 μm with accumulated reduction. The fraction of low-angle boundaries increased while that of high-angle boundaries decreased. This was due to the accumulation of residual stress, which induced stress corrosion in the material interior, further increased the corrosive tendency and contributed to the expansion of pitting corrosion.
- From the TEM and FIB characterizations, it is concluded that corrosion pits expanded along the rolling direction under the interactive effect of pitting corrosion and stress corrosion. Meanwhile, anions aggregated in surface defects, such as dislocations, metastable pits and microcracks, which thereby accelerated the pitting corrosion of the surface.
Point 2: Please match the Rct values from Table 6 with the EIS results from Figure 4a and with the icorr values from Table 5.
Response to point 2: We have recorrected the data and revised the calculation errors in Table 5, Table 6, and matched the Rct, Rpo values from Table 6 with the EIS results from Figure 4a and with the icorr values from Table 5.
Details of the response were seen in the attachment, the revised manuscript were resubmitted.
